# Pt-O bond as an active site superior to Pt$^0$ in hydrogen evolution reaction

Fei-Yang Yu[1,3], Zhong-Ling Lang[1,3], Li-Ying Yin[1], Kun Feng[2], Yu-Jian Xia[2], Hua-Qiao Tan[1]*, Hao-Tian Zhu[1], Jun Zhong[2], Zhen-Hui Kang[2]* & Yang-Guang Li[1]*

The oxidized platinum (Pt) can exhibit better electrocatalytic activity than metallic Pt$^0$ in the hydrogen evolution reaction (HER), which has aroused great interest in exploring the role of oxygen in Pt-based catalysts. Herein, we select two structurally well-defined poly-oxometalates $Na_5[H_3Pt^{(IV)}W_6O_{24}]$ ($PtW_6O_{24}$) and $Na_3K_5[Pt^{(II)}_2(W_5O_{18})_2]$ ($Pt_2(W_5O_{18})_2$) as the platinum oxide model to investigate the HER performance. Electrocatalytic experiments show the mass activities of $PtW_6O_{24}$/C and $Pt_2(W_5O_{18})_2$/C are 20.175 A mg$^{-1}$ and 10.976 A mg$^{-1}$ at 77 mV, respectively, which are better than that of commercial 20% Pt/C (0.398 A mg$^{-1}$). The in situ synchrotron radiation experiments and DFT calculations suggest that the elongated Pt-O bond acts as the active site during the HER process, which can accelerate the coupling of proton and electron and the rapid release of $H_2$. This work complements the knowledge boundary of Pt-based electrocatalytic HER, and suggests another way to update the state-of-the-art electrocatalyst.

[1] Key Laboratory of Polyoxometalate Science of the Ministry of Education, Faculty of Chemistry, Northeast Normal University, Changchun 130024, China. [2] Jiangsu Key Laboratory for Carbon-based Functional Materials and Devices, Institute of Functional Nano and Soft Materials (FUNSOM), Soochow University, Suzhou 215123, China. [3] These authors contributed equally: Fei-Yang Yu, Zhong-Ling Lang. *email: tanhq870@nenu.edu.cn; zhkang@suda.edu.cn; liyg658@nenu.edu.cn

Platinum (Pt) is generally considered as a state-of-the-art electrocatalyst for the hydrogen evolution reaction (HER)[1–8]. In recent decades, enormous efforts have been made to design Pt-based catalysts to boost the utilization and catalytic efficiency of Pt through the composition, morphology, and crystal phase-engineering strategies[9–13]. Most of these studies have revealed the inherent catalytic activity of $Pt^0$ metal[14–23]; however, some interesting phenomena involved in oxidized platinum are still far from being studied. For example, when the metal Pt is oxidized, its HER activity can be obviously better than that of metal element $Pt^0$ (Supplementary Figs. 1 and 2). The reason has not been clearly clarified so far. Recently, great efforts have been made to prepare platinum oxide models such as $PtO_x/TiO_2$[24] and $MoS_x$-O-$PtO_x$[25] and confirmed that the presence of oxygen in Pt catalysts did possess superior HER catalytic performance comparable to that of commercial Pt/C and even superior to that of $MoS_x$-Pt. These pioneering works inspired chemists to find more suitable and distinct platinum oxide models so as to reveal the role of oxygen in Pt-based electrocatalysts and develop new efficient electrocatalysts superior to commercial Pt/C.

Considering that polyoxometalates (POMs) are a unique type of nanoscale metal-oxo clusters with definite structures[26–29] and can be used to simulate the surface of metal oxides[30,31], the Pt-containing POMs could be a readily available and ideal platinum oxide model to investigate the electrocatalytic HER. Therefore, we selected two structurally well-defined Pt-containing POMs $Na_5[H_3Pt^{(IV)}W_6O_{24}]$ (abbr. $PtW_6O_{24}$) and $Na_3K_5[Pt^{(II)}_2(W_5O_{18})_2]$ (abbr. $Pt_2(W_5O_{18})_2$) as model catalysts to investigate their HER performance. Electrochemical experiments show that $PtW_6O_{24}$/C and $Pt_2(W_5O_{18})_2$/C electrocatalysts with 1 wt% Pt content exhibit superior catalytic activities. The overpotentials of $PtW_6O_{24}$/C and $Pt_2(W_5O_{18})_2$/C with 1 wt% Pt content are 22 and 26 mV at 10 mA $cm^{-2}$, and their mass activities are 20.175 and 10.976 A $mg^{-1}$ at an overpotential of 77 mV, respectively, which are better than that of commercial 20% Pt/C (0.398 A $mg^{-1}$). A series of control experiments, in situ synchrotron radiation experiments, and density functional theory (DFT) calculations suggest that the Pt-O bond in POMs should be the active site for HER. Specifically, Pt is mainly an electron-obtaining center, while its coordinated O atoms are proton-capturing centers. During the HER process, when more electrons and protons were injected, the elongated Pt-O bond accelerates the coupling of protons and electrons, which leads to the rapid release of $H_2$ from the Pt-O bond.

## Results

### Structure and electronic property of $Na_5[H_3PtW_6O_{24}]$ ($PtW_6O_{24}$).
$PtW_6O_{24}$ compound was synthesized according to the literature[32]. As depicted in Fig. 1a and Supplementary Figs. 3 and 4, the structure of $PtW_6O_{24}$ is constructed by a $\{PtO_6\}$ octahedron connected to six $\{WO_6\}$ octahedra in an edge-sharing mode. The center Pt atom is surrounded by six O atoms forming the $\{PtO_6\}$ octahedron with a Pt-O bond length of 2.005–2.020 Å. As the pH decreases during synthesis, the number of protons on $PtW_6O_{24}$ cluster increases (Supplementary Table 1). Furthermore, $PtW_6O_{24}$ exhibits excellent stability in a wide pH range (0–6) and under different potentials at room temperature (Supplementary Figs. 5 and 6). The electrospray-ionization mass spectra (EIS) and capillary electrophoresis (CE) show that $PtW_6O_{24}$ compound has reversible redox-active property and good stability (Supplementary Figs. 7 and 8).

X-ray photoelectron spectroscopy (XPS) also shows the definite oxidized platinum feature (Fig. 1 and Supplementary Fig. 9). In the high-resolution XPS spectra of Pt (Fig. 1b), peaks for Pt $4f_{7/2}$ and Pt $4f_{5/2}$ are located at 73.9 and 77.3 eV, respectively, which

are in accordance with the presence of Pt(IV) as reported in the literature. It is noteworthy that there are no signals at 71.4 and 74.7 eV, indicating the absence of metallic Pt[33]. The XPS spectra of O 1$s$ is depicted in Fig. 1c, the peaks at 529.6 and 530.7 eV belong to W=O and W-O-W bonds, respectively. The peaks belong to 532.1 and 533.5 eV can be attributed to the protonation of the Pt-OH-W bond and crystalline $H_2O$, respectively[34]. These experiments further demonstrate that $PtW_6O_{24}$ can be a platinum oxide model to carry out the electrochemical study.

First, the electrochemical cyclic voltammetry (CV) experiments of $PtW_6O_{24}$ were studied in acetonitrile and 0.5 M $H_2SO_4$ aqueous solution, respectively[35,36]. As depicted in Supplementary Fig. 10a, $PtW_6O_{24}$ exhibits a series of obvious redox peaks in the range of −1.5 to 2.0 V in acetonitrile. Specifically, the two redox peaks at 0.68 and 1.02 V are ascribed to the stepwise Pt(III) → Pt(II) and Pt(IV) → Pt(III) processes, respectively (Supplementary Figs. 10–13). The redox peak at −1.2 V is attributed to the reduction of tungsten. The CV results were further simulated by the DFT calculation, which showed that the injection of the initial two electrons mainly occurs on the Pt center (Fig. 1d and Supplementary Fig. 14b), leading to a decrease of the oxidation state from $Pt^{IV}$ to $Pt^{II}$. The DFT calculation further suggests that after $PtW_6O_{24}$ species were 2$e$-reduced, the main contribution to the LUMO became the $d$ orbitals of the surrounding W centers, meaning that the third-reduction step will occur on W instead of Pt. Both CV experiments and DFT calculation reveal that the Pt center in $PtW_6O_{24}$ can more easily obtain electrons than W in $PtW_6O_{24}$, but no metallic $Pt^0$ is generated in the whole electrochemical process. The CV curve of $PtW_6O_{24}$ measured in 0.5 M $H_2SO_4$ is obviously different from that in acetonitrile (Supplementary Fig. 10b). Only the redox peaks of platinum can be detected, and then the hydrogen evolution signal arises and covers the redox peak region of tungsten. Moreover, the molecular electrostatic potential (MEP, see Fig. 1e) maps suggest O atoms on $PtW_6O_{24}$ species should be the main proton-capturing centers (because the red area of O atoms in Fig. 1e represents the most basic positions), which may serve as proton transfer stations to continuously supply $H_2$ generation. All above results imply that $PtW_6O_{24}$ may function as an efficient HER catalyst.

### Electrocatalytic HER performance of $PtW_6O_{24}$/C.
The HER performance of $PtW_6O_{24}$ was assessed by fabricating a $PtW_6O_{24}$/C composite electrocatalyst. Its preparation method and physical characterization are shown in Supplementary Figs. 15–21. The inductively coupled plasma-atom emission spectrometry (ICP-AES) (Supplementary Table 2) demonstrated a 1 wt% Pt content in $PtW_6O_{24}$/C. The electrocatalytic activity of 1% $PtW_6O_{24}$/C for the HER was evaluated and compared to commercial 20% Pt/C in $N_2$-saturated 0.5 M $H_2SO_4$. The polarization curves in Fig. 2a and Supplementary Fig. 22 show that 1% $PtW_6O_{24}$/C exhibits excellent HER activity with over-potentials of 22, 55, and 65 mV at current densities of 10, 70, and 100 mA $cm^{-2}$, respectively, which are better than those of commercial 20% Pt/C (33, 90, and 118 mV) and 1% Pt/C (68, 269, and 357 mV), and exceed most of the reported Pt-based catalysts (Supplementary Fig. 23 and Table 7). As a comparison, series of $PtW_6O_{24}$/C with different loading amount of Pt have been prepared. As described in Supplementary Fig. 24, the HER performance of catalysts was obviously enhanced with the increase of Pt loading. However, when the loading amount of Pt reaches 5%, the electrocatalytic activity did not increase significantly. It is presumed that the partial aggregation of $PtW_6O_{24}$ species results in a decrease in the utilization of the catalyst. In addition, as described in Fig. 2b and Supplementary Fig. 25, the Tafel slope of

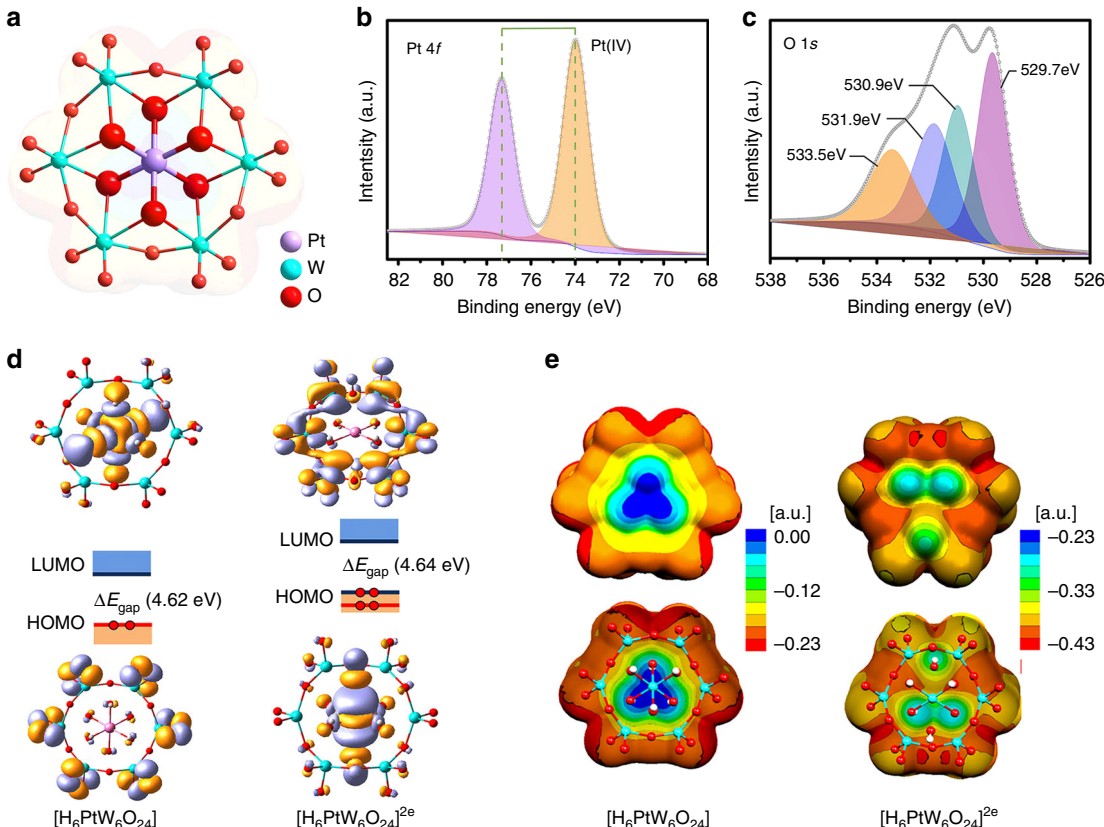

**Fig. 1 Structure and electronic properties of PtW$_6$O$_{24}$. a** The ball and stick representation of PtW$_6$O$_{24}$ with the corresponding semitransparent map of molecular electrostatic potential (MEP). **b**, **c** High-resolution XPS spectra of Pt and O for PtW$_6$O$_{24}$. **d** 3D representation of the highest occupied molecular orbital (HOMO) and the lowest unoccupied molecular orbital (LUMO) for [H$_6$PtW$_6$O$_{24}$] and [H$_6$PtW$_6$O$_{24}$]$^{2e}$. Additional reduced states are given in Supplementary Fig. 14. **e** Full and one-half views of the MEP distribution for [H$_6$PtW$_6$O$_{24}$] and [H$_6$PtW$_6$O$_{24}$]$^{2e}$: the red and blue or green identify more and less nucleophilic regions for H$^+$. Supplementary Table 6 discusses the proton distributions.

1% PtW$_6$O$_{24}$/C is 29.8 mV dec$^{-1}$, which is consistent with the Volmer–Tafel mechanism, in which the recombination of chemisorbed H atoms and ions is the rate-determining step[37]. Furthermore, the 1% PtW$_6$O$_{24}$/C exhibits an exchange current density of 1.65 mA cm$^{-2}$ (Supplementary Fig. 26), which means a superior intrinsic electrocatalytic activity. The mass activity and specific activity were normalized by the mass loading and the electrochemical surface area (ECSA) of Pt. As depicted in Fig. 2c, at an overpotential of 77 mV, PtW$_6$O$_{24}$/C displays a mass activity of 20.175 A mg$^{-1}$, while the mass activity of 20% Pt/C is 0.398 A mg$^{-1}$. Furthermore, 1% PtW$_6$O$_{24}$/C displays a specific activity of 35.266 mA cm$^{-2}$ at 50 mV, and the value of 20% Pt/C is 0.132 mA cm$^{-2}$ under the same condition. As shown in Supplementary Fig. 27, the turnover frequencies (TOFs) displays a near linear increase with the overpotential. At an overpotential of 100 mV, 1% PtW$_6$O$_{24}$/C exhibits a high TOFs of 33.35 s$^{-1}$, which is a 58.5-fold increase over 20% Pt/C (0.57 s$^{-1}$). Figure 2d indicates the relationship between the current density, the overpotential and the Pt content, demonstrating that the current density of 1% PtW$_6$O$_{24}$/C is obviously better than that of 20% Pt/C at all overpotentials. Even compared with Pt/C with a higher Pt content, its value is also better in most of the overpotentials.

Electrochemical impedance spectroscopy (EIS) are depicted in Supplementary Figs. 28 and 29 and suggest that 1% PtW$_6$O$_{24}$/C possesses an extremely low charge transfer resistance ($R_{ct} = 6.7\ \Omega$), which is obviously lower than that of 20% Pt/C ($R_{ct} = 46.8\ \Omega$), demonstrating the fast Faradaic process between the interface of the catalysts and electrolyte. The Faradic efficiency of 1% PtW$_6$O$_{24}$/C is nearly 100% for the HER, resulting in the high

evolution efficiency of H$_2$ (Supplementary Fig. 30). As shown in Supplementary Fig. 31 (anti-toxicity test), the activity of 1% PtW$_6$O$_{24}$/C exhibits negligible changes in the presence of Co$^{2+}$, Fe$^{2+}$, Mn$^{2+}$, and Ni$^{2+}$ ions, while the performance of 20% Pt/C decreases obviously after three cycles. Such results indicate that 1% PtW$_6$O$_{24}$/C possesses good anti-toxicity properties.

Besides the aforementioned features, stability is also an important factor for evaluating an excellent electrocatalyst. The accelerated degradation test (ADT) in Fig. 2e was used to estimate the electrocatalytic durability. After 1000 and 3000 cycles, the polarization curves of PtW$_6$O$_{24}$/C show a slight loss. In addition, the long-term stability test for PtW$_6$O$_{24}$/C was carried out at an overpotential of 30 mV for 24 h (Fig. 2e, inset). The current density exhibits a small loss, demonstrating the good electrocatalytic stability of PtW$_6$O$_{24}$/C. The Pt content in PtW$_6$O$_{24}$/C shows negligible loss before and after the HER, indicating no dissolution of the catalyst during the electrocatalytic process (Supplementary Table 3). The Transmission electron microscopy (TEM) images of PtW$_6$O$_{24}$/C after long-term electrochemical test demonstrate that its morphology stays the same without aggregation, meaning the good stability of PtW$_6$O$_{24}$/C (Supplementary Fig. 32). Infrared (IR) spectra of 1% PtW$_6$O$_{24}$/C after long-term electrocatalytic tests also suggest that 1% PtW$_6$O$_{24}$/C is stable during the electrochemical reaction process (Supplementary Fig. 33).

**Electrocatalytic mechanism of PtW$_6$O$_{24}$/C.** First, a series of control experiments were studied to understand the origins of the excellent HER activities of PtW$_6$O$_{24}$/C. All the reference catalysts

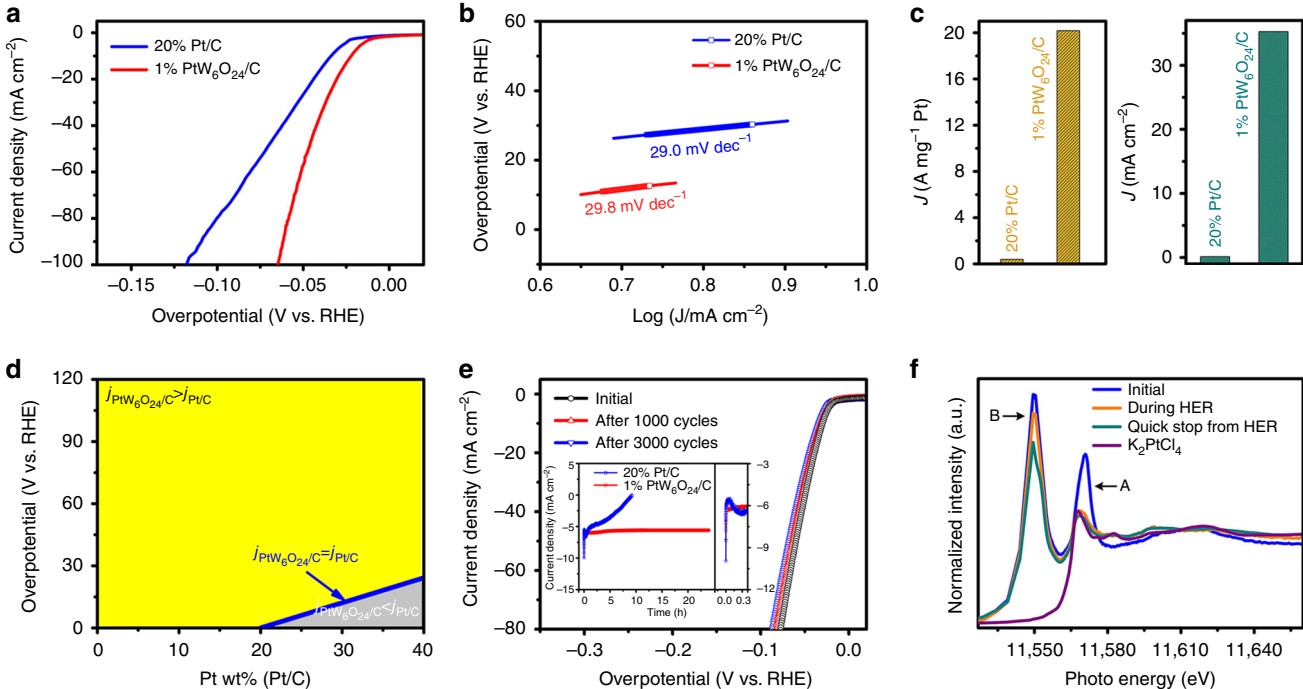

**Fig. 2 HER performance of the PtW$_6$O$_{24}$ catalyst. a** The polarization curves of 1% PtW$_6$O$_{24}$/C and 20% Pt/C in N$_2$-saturated 0.5 M H$_2$SO$_4$. **b** Tafel slope of 1% PtW$_6$O$_{24}$/C and 20% Pt/C. **c** Mass activity of PtW$_6$O$_{24}$/C and 20% Pt/C at 77 mV (left), the specific activity of 1% PtW$_6$O$_{24}$/C and 20% Pt/C at 50 mV (right). **d** The relationship between the current density, the overpotential, and the Pt content. The blue area indicates that the current density of PtW$_6$O$_{24}$/C and 20% Pt/C is the same. The yellow area indicates that the current density of PtW$_6$O$_{24}$/C is greater than Pt/C, and the gray area indicates that the current density of PtW$_6$O$_{24}$/C is lower than Pt/C. **e** The polarization curves of 1% PtW$_6$O$_{24}$/C before and after 1000 and 3000 cycles at scan rate of 5 mV s$^{-1}$. Inset: Time-dependent current density current of 1% PtW$_6$O$_{24}$/C and 20% Pt/C within 24 h (left). The locally enlarged plot of the time-dependent current density current for the first 0.5 h (right). **f** In situ XANES spectra of the 1% PtW$_6$O$_{24}$/C sample at Pt $L_3$-edge. Feature A is attributed to Pt $L_3$-edge and feature B is attributed to W $L_2$-edge.

such as LaW$_{10}$/C, Kenjet black, and Pt(C$_2$H$_5$N$_4$O$_2$)$_2$/C exhibit poor HER activities. These results indicate that PtW$_6$O$_{24}$/C has excellent HER catalytic properties due to the existence of {PtO$_6$} core (Supplementary Fig. 34).

Furthermore, in situ X-ray adsorption spectroscopy (XAS) was performed under pretreatment and electrocatalytic conditions to gain insight into the electronic state variation of Pt. As shown in Fig. 2f, in initial (before HER) state, the Pt K-edge X-ray absorption near edge structure (XANES) spectrum of PtW$_6$O$_{24}$/C show similar intensity to the Na$_2$Pt(OH)$_6$ reference, manifesting that the Pt element in PtW$_6$O$_{24}$/C takes on a similar valence state as in Na$_2$Pt(OH)$_6$. On the other hand, a shift in the edge position to lower energy was observed when moved to HER condition, and the intensity for Pt was significantly decreased. This indicates a reduction in the Pt oxidation state. In addition, the XAS during HER was found closely matching with K$_2$PtCl$_4$ species, strongly suggesting the existence of Pt$^{II}$ electronic state during the HER catalytic process. These results demonstrate that the mechanism of the HER process could be attributed to the formation of the Pt$^{II}$ intermediate. The valence state of Pt decreased from Pt(IV) to Pt (II), and then the state was maintained between Pt(II) and Pt(I), which is consistent with the results of CV tests. Feature B is due to the interference of W $L_2$-edge, which does not affect the analysis of main peak of PtW$_6$O$_{24}$/C. During the HER process, the intensity of feature B also decreased, indicating that W might also participate in the HER reaction. This result indicates that no metallic Pt$^0$ is formed during the electrocatalytic HER process. The Pt-O bonds always exist in the PtW$_6$O$_{24}$/C catalyst, except for the valence state changes of Pt. This result is consistent with the above experimental data, as well as the previous literatures[35,36].

DFT calculations were employed at the M06 level to survey the detailed pathways of H$_2$ generation (Fig. 3). [H$_6$PtW$_6$O$_{24}$] can engage in multiple reduction and protonation reactions under electrocatalytic conditions (Supplementary Table 6). However, H$_2$ evolution from the 2e-reduction state ([H$_6$PtW$_6$O$_{24}$]$^{2e/2H}$) is completely restricted due to the high energy demand of 2.41 eV (Supplementary Fig. 35c). Among the considered routes, the most accessible catalytic cycle is proposed to be through the highly reduced [H$_6$PtW$_6$O$_{24}$]$^{4e/4H}$ state to generate H$_2$ and then regenerate [H$_6$PtW$_6$O$_{24}$]$^{2e/2H}$ (Fig. 3a). Noticeably, two Pt-O bonds are apparently weakened (or elongated) by the two-electron reduction; therefore, providing an available site for H attack (Supplementary Fig. 14) to form a Pt-H state. The configuration with one H on the Pt site ([H$_6$PtW$_6$O$_{24}$]$^{4e/4H(Pt)}$) was revealed to be 0.04 eV energetically more favorable than on the O ([H$_6$PtW$_6$O$_{24}$]$^{4e/4H(O)}$) due to a fast intramolecular electronic rearrangement process (Supplementary Figs. 35b and 36). Starting from the [H$_6$PtW$_6$O$_{24}$]$^{4e/4H(Pt)}$ intermediate, a transition state (TS) search demonstrated that H$_2$ production is kinetically promising with a barrier of only 0.15 eV, and the singlet-state [H$_6$PtW$_6$O$_{24}$]$^{2e/2H}$ can be spontaneously reconstituted via an exothermic process (Fig. 3b). Our calculations assign a partial charge of $Q_H = 0.12$ to the H in the Pt-H moiety (clearly smaller than that on the OH moiety $Q_H = 0.47$), supplying the active H and combining with adjacent proton from O to encourage H$_2$ evolution (Fig. 3b, inset). In addition, the catalytic cycle is computed to be catalytically efficient between the Pt(II) and Pt(I)-involved intermediates (Supplementary Fig. 36), and these charge assignments are consistent with the trends obtained in XAS.

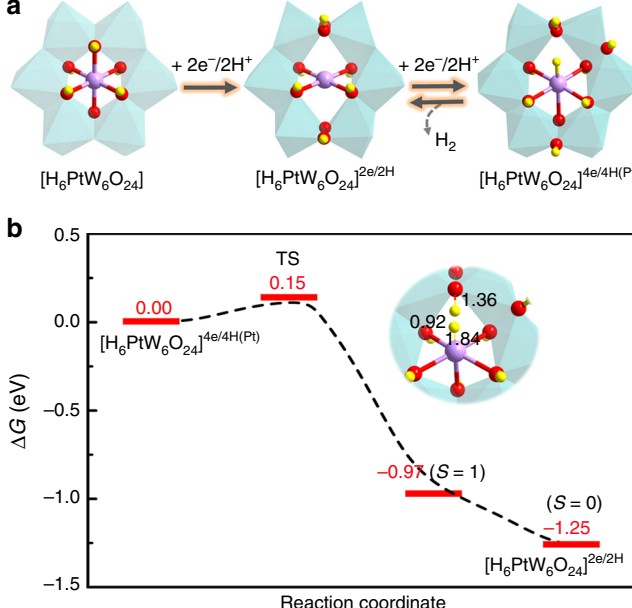

**Fig. 3 H₂ evolution pathways over [PtW₆O₂₄]. a** Mechanistic scheme of the HER catalyzed by PtW₆O₂₄/C. [H₆PtW₆O₂₄] experiences a two-electron/proton-coupled reduction to form [H₆PtW₆O₂₄]$^{2e/2H}$ (Pt$^{II}$), agreeing well with the intermediate detected by XAS. Two further reductions and intramolecular electronic recombination (Supplementary Fig. 35) were suggested to generate the active [H₆PtW₆O₂₄]$^{4e/4H(Pt)}$. Catalytic H₂ formation is proposed to occur between [H₆PtW₆O₂₄]$^{2e/2H}$ and [H₆PtW₆O₂₄]$^{4e/4H}$. Other pathways are discussed in Supplementary Fig. 35c. **b** Free energy diagrams for H₂ production with an extremely low barrier. H is colored with yellow for clarity.

## Discussion

Based on the DFT and XAS results, [H₆Pt$^{(II)}$W₆O₂₄]$^{2e/2H}$ may represent an important intermediate for the high HER performance of H₆PtW₆O₂₄. This result aroused our curiosity to detect the electrocatalytic activity of another POM Na₃K₅[Pt$^{(II)}$₂(W₅O₁₈)₂] (Pt₂(W₅O₁₈)₂), since it contains a similar Pt$^{II}$-O moiety in the molecular structure. The overpotential of 1% Pt₂(W₅O₁₈)₂/C is 22 mV at 10 mA cm$^{-2}$. Its exchange current density and mass activity at 77 mV are 1.65 mA cm$^{-2}$ and 20.175 A mg$^{-1}$, respectively, which are quite close to PtW₆O₂₄. More detailed data are provided in the Supplementary Figs. 37–57 and Supplementary Tables 4 and 5)[38]. This result further confirms that Pt-O bond is the active site during the HER process. Herein, it should be also clarified that although commercial Pt/C is widely used as a standard reference for HER research, its performance is already not the best. No matter in size, morphology, and dispersion of metal Pt, there exists enough space to improve its catalytic activity[9–14,39]. Thus, surpassing commercial Pt/C does not mean that metal Pt-based catalysts are out of date, which exactly suggests an important driving force for deeply developing such state-of-the-art catalysts.

In summary, we selected two structurally well-defined Pt-containing POMs as the platinum oxide model to reveal the role of O atom in Pt-based electrocatalysts towards HER. The electrochemical experiments show that PtW₆O₂₄/C and Pt₂(W₅O₁₈)₂/C possess the overpotentials of 22 and 26 mV at a current density of 10 mA cm$^{-2}$, and their mass activities are 20.175 and 10.976 A mg$^{-1}$, respectively. In situ XAS experiments and DFT calculations suggest that the Pt-O bond should be the active site for the HER. Specifically, Pt is mainly an electron-obtaining center, while the O acts as the proton-adsorption center. When

extra electrons and protons were injected during electrochemical process, the elongated Pt-O site accelerates the coupling of electron and proton and leads to a rapid release of H₂ on the Pt-O bond. Therefore, Pt-O can be utilized as a new active site towards HER. This work answers the important role of O atoms in the oxidized platinum-based electrocatalytic HER, which may bring an another enlightenment for the design and preparation of more efficient Pt-based electrocatalysts in the near future.

## Methods

**Characterization.** Single-crystal X-ray diffraction data for Pt-POMs was collected by using a Bruker Smart Apex CCD diffractometer with Mo-Kα radiation ($\lambda$ = 0.71073 Å) at the temperature of 298(2) K. Powder X-ray diffraction measurements were carried out on a Rigaku D/max-IIB X-ray diffractometer with Cu-Kα radiation ($\lambda$ = 1.5418 Å). TEM and high-resolution TEM images were obtained on JEOL-2100F and JEM-F 200 instruments at an accelerating voltage of 200 kV. Scanning transmission electron microscopy (STEM) images were obtained on a Titan Cubed Themis G2 300 equipped with a probe corrector and HF5000. The XPS measurements were performed on a KRATOS Axis ultra DLD X-ray photoelectron spectrometer with a monochromatized Mg Kα X-ray source ($h\nu$ = 1283.3 eV). XANES and extended X-ray absorption fine structure (EXAFS) data were collected on the BL14W beamline at the Shanghai Synchrotron Radiation Facility, operated at 3.5 GeV with injection currents of 140–210 mA. The ICP-AES elemental analyses were performed on a Teledyne Leeman Labs ICP-AES spectrometer. Electrochemical measurements and electrocatalytic HER performance were tested by using a CHI760E workstation (CH Instruments, China). The evolved gases during HER decomposition were detected by gas chromatography (Shimadzu, GC-2014C with a thermal conductivity detector). K₂PtCl₄, Na₂Pt(OH)₆, Na₂WO₄·2H₂O, La(NO₃)₃·6H₂O, and Ketjen black carbon were purchased from Aladdin Industrial Co., Ltd. Nafion solution (5 wt%) and commercial 40% Pt/C, 20% Pt/C, and 5% Pt/C were purchased from Alfa Aesar China (Tianjin) Co., Ltd. All solution used in experiments were prepared with Millipore water (18.2 MΩ).

**Synthesis of PtW₆O₂₄/C.** Crystal (0.065 g) of Na₅[H₃Pt(IV)W₆O₂₄] was uniformly dispersed in 1 mL H₂O, and 5 mg Ketjen black carbon was added to, stirring at room temperature for 2 h. Then, 10 μL melamine-formaldehyde was added to the aqueous and stirred 4 h. The electrocatalyst can be obtained after centrifuged and dried. The obtained sample is denoted as PtW₆O₂₄/C.

**Synthesis of Pt₂(W₅O₁₈)₂/C.** Crystal (0.065 g) of Na₃K₅[Pt(II)₂(W₅O₁₈)₂] was uniformly dispersed in 1 mL H₂O, and 5 mg Ketjen black carbon was added to, stirring at room temperature for 2 h. Then, 10 μL melamine-formaldehyde was added to the aqueous and stirred 4 h. The electrocatalyst can be obtained after centrifuged and dried. The obtained sample is denoted as Pt₂(W₅O₁₈)₂/C.

**Electrochemical characterization of Pt-POMs.** All reagents are guaranteed reagent and chemicals are of high-purity grade, which were purchased from Aladdin Industrial Co., Ltd. The electrolyte was 0.05 M tetrabutylammonium perchlorate/CH₃CN deoxygenated thoroughly for 30 min with pure nitrogen and keep under a positive pressure of this gas during the electrochemical tests. The working electrode was well clean bare glassy carbon. The platinum wire was used as the counter electrode. Non-aqueous Ag/Ag⁺ electrode served as reference electrode. The filling solution of non-aqueous Ag/Ag⁺ electrode was 0.01 M AgNO₃/CH₃CN. All electrochemical tests were carried out by using a CHI760E workstation and performed at room temperature under atmospheric pressure.

**Electrochemial measurements of HER.** HER tests were carried out a conventional three-electrode electrochemical system in N₂-saturated 0.5 M H₂SO₄ at 300 K. A modified glassy carbon electrode (GCE; $d$ = 3 mm) was used as the working electrode. A saturated calmoel electrode (SCE) and a carbon rod served as reference electrode and counter electrode, respectively. All electrochemical tests were carried out by using a CHI760E workstation. Polarization curves were tested at a scan rate of 5 mV s$^{-1}$ in 0.5 M H₂SO₄. The measured potentials vs. SCE were standardized with a reversible hydrogen electrode (RHE) based on $E$ vs. RHE = $E^{\theta}_{SCE}$ + 0.059 pH ($E^{\theta}_{SCE}$ = 0.242 V). All data are presented with IR compensation. the electrochemical double layer capacitance ($C_{dl}$) was evaluated by cyclic voltammogram (CV) from 0.1 to −0.1 V with different scan rates. The ADT was measured by CV with sweeps at 100 mV at between +0.1 and −0.2 V vs. RHE. The long-term stability was measured at controlled potential.

**DFT computational details.** All calculations were performed through the facilities provided by the Gaussian09 package (revision D.01)[40]. Geometry optimizations for all intermediates and TSs were carried out at the M06 level without symmetry restrictions[41]. The LANL2DZ basis set was employed for the Pt and W, whereas the 6–31G(d, p) basis set was used for the O and H[42,43]. To confirm the stability of all structures, frequency calculations were performed at the same level as optimization.

The TSs were confirmed by the existence of only one imaginary frequency along the reaction coordinate and intrinsic reaction coordinate (IRC) calculations, which indeed connect the right reactants and products (Supplementary Fig. S56)[44]. The solvation effects of water were introduced by using the PCM model[45]. Furthermore, the single-point energies of all stationary points were completed at (U)M06/PCM(H$_2$O)/[6-311++G(d,p)/SDD(Pt&W)] level for all energy calculations[46]. Optimized coordinates (*xyz*) for all related species are performed Supplementary Table 8. Finally, a data set of computational results is available in the ioChem-BD repository and can be accessed via https://doi.org/10.19061/iochem-bd-6-27 (http://www.iochembd.org/)[47].

## Data availability

All relevant data are available from the corresponding author on request.

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

## Acknowledgements

This work is financially supported by the National Natural Science Foundation of China (grant nos. 21671036, 21771033, 51725204, 21771132, 21471106, 51972216), National MCF Energy R&D Program (2018YFE0306105), the Fundamental Research Funds for the Central Universities (grant nos. 2412018BJ001, 2412018ZD007, and 2412018QD005), the Opening Project of Key Laboratory of Polyoxometalate Science of the Ministry of Education (grant no. 130014556), the China Postdoctoral Science Foundation funded project (grant no. 2018M631849), the Foundation of Jilin Educational Committee (grant no. JJKH20190268KJ), and the Scientific Development Project of Jilin Province (grant no. 20190201206JC). The Natural Science Foundation of Jiangsu Province (BK20190041, BK20190828), Guangdong Province Key Area R&D Program (2019B010933001), Collaborative Innovation Center of Suzhou Nano Science and Technology, the Priority Academic Program Development of Jiangsu Higher Education Institutions (PAPD), and the 111 Project. This article is dedicated to the memory of our most beloved supervisor Professor En-Bo Wang, who passed away on 28 September 2019.

## Author contributions

Y.-G.L., Z.-H.K., and H.Q.T. conceived the whole experiments. F.Y.Y. performed the main experiments. Z.L.Z. performed all the DFT calculations. L.Y.Y. performed the CE experiments. K.F., Y.J.X. and J.Z. performed the in situ XAS experiments. H.-T.Z. solved the crystals. Y.-G.L. and H.-Q.T. polished the whole paper. All authors discussed the results and implications and commented on the manuscript at all stages.

## Competing interests

The authors declare no competing interests.
