## [Peer Review File · Nature Communications]

Reviewers' comments:

Reviewer #1 (Remarks to the Author):

The authors convincingly show that (in acetonitrile) Pt(IV) in PtW₆O₂₄ is reduced to Pt(II) after which the third electron reduces W(VI). As such, catalysis in aqueous acid is reasonably due to Pt(II) (and perhaps Pt(I) as shown later), rather than to Pt(0). This is also consistent with reductions of numerous other transition-metal (TM) substituted heteropolytungstates, in which reduction of bound TM cations to their metallic state does not occur due to more favorable reduction of W(VI). Stability to repeated cycling and resistance to poisoning by added TM cations (referred to by the authors as "anti-toxicity") further support the presence of a redox-active molecular catalyst, as opposed to the reductive degradation of the complexes and formation of Pt(0) clusters. Findings from DFT calculations argue that the kinetically competent species responsible for H₂ evolution actually involves reduction of the W(VI)-based LUMO by two electrons. This is not only reasonable, but quite interesting in its implications for further catalyst design. Notably, quite similar results are obtained for the Pt(II) complex (Pt₂(W₅O₁₈)₂). In summary, the key finding is that the partial reduction of Pt cations bound via oxide ligands to reduced W ions within molecular heteropolytungstate clusters are more effective than Pt(0), and comparable to some of the most effective reported in the literature (wrt overpotential at 10 mA cm⁻¹) in the hydrogen evolution reaction. Moreover, the W-based clusters appear to serve as stable, redox catalysts, consistent with reversible reduction / reoxidation under turnover conditions.

This an interesting, clearly presented, high-quality and significant work, potentially suitable for publication in Nature Comm.

Before publication, the authors should provide definitive evidence that the molecular complexes are indeed stable under turnover conditions. One way to do this would be to run bulk-electrolytic H₂ productions reactions using solutions of the complexes in sulfuric acid (as done to obtain the CV data in water). After a reasonable number of turnovers, the complexes in the solution can be characterized, for example by electrospray-ionization mass spectroscopy (perhaps after precipitation from water using quaternary ammonium cation salts, and redissolution in MeCN). It might be necessary to first oxidize the reduced complexes by addition of Br₂, to avoid electron transfer to O₂ and the formation of hydroxide (which could cause hydrolytic degradation). This simple experiment would add an intriguing dimension to the work, and increase its generality and significance. Other approaches would also be acceptable; the main point is to demonstrate that the complexes behave as reversible redox-active catalysts.

Ira A. Weinstock

Reviewer #2 (Remarks to the Author):

Revision after more data is provided.

This reviewer would like to get access to the DFT computed geometries and frequencies of all species included in the manuscript, particularly those corresponding to the Tafel-like H-H forming transition state. The barrier reported is very low, and has to be verified. Normally, this data is included in the Supplementary Information section, but it is missing here. Alternatively, the authors may upload input and output files to any digital repository and provide links to access the data.

Minor questions:

The version of the program used needs to be properly mentioned, as well as properly cited.

Reviewer #3 (Remarks to the Author):

The manuscript submitted by Li et al report on the remarkable HER efficiency of two platinum-containing polyoxometalates (POMs), abbreviated PtW6O24 and Pt2(W5O18)2. In a general point of view, the presented work appears well-done and highly convincing. The authors demonstrate that containing-platinum POMs as soluble oxide analogues are able to mimic the platinum oxide behavior toward hydrogen evolution giving highly efficient HER catalysts. The manuscript proposes a multi-scale characterization of the catalyst using electrochemistry, XPS, X-ray absorption spectroscopy (XANES and EXAFS), TEM and STEM, vibrational spectroscopies (Raman and infrared, impedance spectroscopy.... Furthermore, experimental data were supported by theoretical calculation at the DFT level, thus giving a set of reliable arguments consistent with the hypotheses.

Undoubtedly, these two Pt-containing POMs are stable, highly processable, very efficient and resistant against usual Pt-contaminant. This submitted manuscript could display the requested scientific quality to be published in NatureCOMM. The first critical point of the MS is certainly the repeated comparison of the HER performances of Pt-POM/catalysts with those of the commercial 20% Pt/C. The sentence "better than that of commercial 20% Pt/C" appears at least twelve times in the text. The reader understand that the commercial 20% Pt/C is a very bad HER catalyst, with not optimized Pt-dispersion and probably containing platinum-contaminant agents. Furthermore, the procedures of preparation of both type of catalyst are necessarily different. For instance, the nature of the used carbon is different, such as Ketjen carbon for Pt-POM/catalysts leading to important changes of the surface area and conductivity. Actually, this is easy to get a poorly HER active platinum-based catalyst. This point must be corrected and the highest HER performances of the two Pt-POMs/C should be commented objectively and scientifically.

Furthermore, the paper is divided in two redundancy part dealing with the HER properties of the PtW6O24/C, and those of the Pt2(W5O18)2/C, respectively. The first part appears very interesting, dealing with the electrochemical behavior of the Pt(IV) center embedded within the {W6O24} framework. Calculations are consistent with the expected quasi-square Pt(II) center, while the HER process resulted from the presence of Pt(I)-H hydrid group. The second part (Pt2(W5O18)2/C) brings nothing new and should be deleted. The authors did not give any arguments about the selection of this Pt2(W5O18)2 POM species.

Less minor point : In the "Methods" section, the synthetic procedure of the two Pt-POMs/C, given in the text is not consistent with the scheme 1 (supp Info). In the text Nafion is used while in the supp Info, Melamine-Formaldehyde is added. This should be cleared.

In conclusion, this paper should disserve publication in NatureCOMM, therefore some revisions are still needed.

Response to reviewers' comments

Reviewer 1

The authors convincingly show that (in acetonitrile) Pt(IV) in $\text{PtW}_6\text{O}_{24}$ is reduced to Pt(II) after which the third electron reduces W(VI). As such, catalysis in aqueous acid is reasonably due to Pt(II) (and perhaps Pt(I) as shown later), rather than to Pt(0). This is also consistent with reductions of numerous other transition-metal (TM) substituted heteropolytungstates, in which reduction of bound TM cations to their metallic state does not occur due to more favorable reduction of W(VI). Stability to repeated cycling and resistance to poisoning by added TM cations (referred to by the authors as "anti-toxicity") further support the presence of a redox-active molecular catalyst, as opposed to the reductive degradation of the complexes and formation of Pt(0) clusters. Findings from DFT calculations argue that the kinetically competent species responsible for H_2 evolution actually involves reduction of the W(VI)-based LUMO by two electrons. This is not only reasonable, but quite interesting in its implications for further catalyst design. Notably, quite similar results are obtained for the Pt(II) complex ($\text{Pt}_2(\text{W}_5\text{O}_{18})_2$). In summary, the key finding is that the partial reduction of Pt cations bound via oxide ligands to reduced W ions within molecular heteropolytungstate clusters are more effective than Pt(0), and comparable to some of the most effective reported in the literature (wrt overpotential at 10 mA cm^{-2}) in the hydrogen evolution reaction. Moreover, the W-based clusters appear to serve as stable, redox catalysts, consistent with reversible reduction / reoxidation under turnover conditions.

This an interesting, clearly presented, high-quality and significant work, potentially suitable for publication in Nature Comm.

Our response:

We do thank your positive comments.

Comment 1: Before publication, the authors should provide definitive evidence that the molecular complexes are indeed stable under turnover conditions. One way to do this would be to run bulk-electrolytic H_2 productions reactions using solutions of the complexes in sulfuric acid (as done to obtain the CV data in water). After a reasonable number of turnovers, the complexes in the solution can be characterized, for example by electrospray-ionization mass spectroscopy (perhaps after precipitation from water using quaternary ammonium cation salts, and redissolution in MeCN). It might be necessary to first oxidize the reduced complexes by addition of Br_2 , to avoid electron transfer to O_2 and the formation of hydroxide (which could cause hydrolytic degradation). This simple experiment would an intriguing dimension to the work, and increase it's generality and

significance. Other approaches would also be acceptable; the main point is to demonstrate that the complexes behave as reversible redox-active catalysts.

Our response:

According to the suggestion, the electrospray-ionization mass spectra (EIS) of $\text{PtW}_6\text{O}_{22}$ compound (before and after HER in 0.5 M H_2SO_4 aqueous solution as well as its re-oxidized species by Br_2) were measured by precipitating the polyoxoanion with tetrabutylammonium (TBA) bromide and dissolved in CH_3CN (see Supplementary Figure 7). Before HER of $\text{PtW}_6\text{O}_{24}$ catalyst, the signals of $m/z = 1689$ can be assigned to $\text{H}_4[\text{Pt}^{\text{IV}}\text{W}_6\text{O}_{24}\text{H}_3]^-$ species (Figure 7 black line). After $\text{PtW}_6\text{O}_{24}$ catalyst has undergone HER for 10 cycles in 0.5 M H_2SO_4 aqueous solution, the signals of $m/z = 1690$, $m/z = 1795$, $m/z = 1898$, and $m/z = 1972$ can be assigned to $\text{H}_4[\text{Pt}^{\text{II}}\text{W}_6\text{O}_{24}\text{H}_5]^-$, $\text{NaH}_3[\text{Pt}^{\text{II}}\text{W}_6\text{O}_{24}\text{H}_5]^- \cdot (\text{CH}_3\text{CN})_2$, $\text{Na}_4[\text{Pt}^{\text{II}}\text{W}_6\text{O}_{24}\text{H}_5]^- \cdot (\text{CH}_3\text{CN})_2(\text{H}_2\text{O})_2$, and $(\text{TBA})\text{NaH}_2[\text{Pt}^{\text{II}}\text{W}_6\text{O}_{24}\text{H}_5]^- \cdot \text{CH}_3\text{CN}$ species, respectively (Figure 7 red line). When the molecular catalyst was re-oxidized by Br_2 in above solution, the signals of $m/z = 1710$, $m/z = 1732$, $m/z = 1753$, $m/z = 1777$ and $m/z = 1798$ can be assigned to $\text{NaH}_3[\text{Pt}^{\text{IV}}\text{W}_6\text{O}_{24}\text{H}_3]^-$, $\text{Na}_2\text{H}_2[\text{Pt}^{\text{IV}}\text{W}_6\text{O}_{24}\text{H}_3]^-$, $\text{NaH}_3[\text{Pt}^{\text{IV}}\text{W}_6\text{O}_{24}\text{H}_3]^- \cdot \text{CH}_3\text{CN}$, $\text{Na}_4[\text{Pt}^{\text{IV}}\text{W}_6\text{O}_{24}\text{H}_3]^-$, and $\text{Na}_3\text{H}[\text{Pt}^{\text{IV}}\text{W}_6\text{O}_{24}\text{H}_3]^- \cdot \text{CH}_3\text{CN}$, respectively. The above experiments demonstrate that $\text{PtW}_6\text{O}_{24}$ compound behaves as a reversible redox-active catalyst (see Supplementary Figure 7 and Table R1-R3).

Supplementary Figure 7 | The electrospray-ionization mass spectra (EIS) of $\text{PtW}_6\text{O}_{24}$ compound before and after HER in 0.5 M H_2SO_4 aqueous solution as well as its re-oxidized species by Br_2 . The test samples are prepared by precipitating the polyoxoanion with tetrabutylammonium (TBA) bromide and dissolved in CH_3CN . (a) Before HER, the signals of $m/z = 1689$ can be assigned to $\text{H}_4[\text{Pt}^{\text{IV}}\text{W}_6\text{O}_{24}\text{H}_3]^-$ species. (b) After $\text{PtW}_6\text{O}_{24}$ catalyst has undergone HER for 10 cycles in 0.5 M H_2SO_4 aqueous solution, the signals of $m/z = 1690$, $m/z = 1795$, $m/z = 1898$, and $m/z = 1972$ can be assigned to $\text{H}_4[\text{Pt}^{\text{II}}\text{W}_6\text{O}_{24}\text{H}_5]^-$, $\text{NaH}_3[\text{Pt}^{\text{II}}\text{W}_6\text{O}_{24}\text{H}_5]^- \cdot (\text{CH}_3\text{CN})_2$, $\text{Na}_4[\text{Pt}^{\text{II}}\text{W}_6\text{O}_{24}\text{H}_5]^- \cdot (\text{CH}_3\text{CN})_2(\text{H}_2\text{O})_2$, and $(\text{TBA})\text{NaH}_2[\text{Pt}^{\text{II}}\text{W}_6\text{O}_{24}\text{H}_5]^- \cdot \text{CH}_3\text{CN}$ species, respectively. (c) When the molecular catalyst was re-oxidized by Br_2 in above solution, the signals of $m/z = 1710$, $m/z = 1732$, $m/z = 1753$, $m/z = 1777$

and $m/z = 1798$ can be assigned to $\text{NaH}_3[\text{Pt}^{\text{IV}}\text{W}_6\text{O}_{24}\text{H}_3]^-$, $\text{Na}_2\text{H}_2[\text{Pt}^{\text{IV}}\text{W}_6\text{O}_{24}\text{H}_3]^-$, $\text{NaH}_3[\text{Pt}^{\text{IV}}\text{W}_6\text{O}_{24}\text{H}_3]^- \cdot \text{CH}_3\text{CN}$, $\text{Na}_4[\text{Pt}^{\text{IV}}\text{W}_6\text{O}_{24}\text{H}_3]^-$, and $\text{Na}_3\text{H}[\text{Pt}^{\text{IV}}\text{W}_6\text{O}_{24}\text{H}_3]^- \cdot \text{CH}_3\text{CN}$, respectively. The above experiments demonstrate that $\text{PtW}_6\text{O}_{24}$ compound behaves as a reversible redox-active catalyst.

Table R1. EIS data of $\text{PtW}_6\text{O}_{24}$ compound before HER

z	m/z (Obs)	m/z (Cal)	Assignment
-1	1689.42	1689.12	$\text{H}_4[\text{Pt}^{\text{IV}}\text{W}_6\text{O}_{24}\text{H}_3]$

Table R2. EIS data of $\text{PtW}_6\text{O}_{24}$ compound after HER

z	m/z (Obs)	m/z (Cal)	Assignment
-1	1690.72	1691.12	$\text{H}_4[\text{Pt}^{\text{II}}\text{W}_6\text{O}_{24}\text{H}_5]$
-1	1794.83	1795.21	$\text{NaH}_3[\text{Pt}^{\text{II}}\text{W}_6\text{O}_{24}\text{H}_5] \cdot (\text{CH}_3\text{CN})_2$
-1	1898.73	1897.22	$\text{Na}_4[\text{Pt}^{\text{II}}\text{W}_6\text{O}_{24}\text{H}_5] \cdot (\text{CH}_3\text{CN})_2(\text{H}_2\text{O})_2$
-1	1971.09	1972.38	$(\text{TBA})\text{NaH}_2[\text{Pt}^{\text{II}}\text{W}_6\text{O}_{24}\text{H}_5] \cdot \text{H}_2\text{O}$

Table R3. EIS data of re-oxidized $\text{PtW}_6\text{O}_{24}$ compound after HER

z	m/z (Obs)	m/z (Cal)	Assignment
-1	1710.27	1711.11	$\text{NaH}_3[\text{Pt}^{\text{IV}}\text{W}_6\text{O}_{24}\text{H}_3]$
-1	1732.31	1733.12	$\text{Na}_2\text{H}_2[\text{Pt}^{\text{IV}}\text{W}_6\text{O}_{24}\text{H}_3]$
-1	1753.30	1752.17	$\text{NaH}_3[\text{Pt}^{\text{IV}}\text{W}_6\text{O}_{24}\text{H}_3] \cdot \text{CH}_3\text{CN}$
-1	1777.35	1777.12	$\text{Na}_4[\text{Pt}^{\text{IV}}\text{W}_6\text{O}_{24}\text{H}_3]$
-1	1798.16	1796.17	$\text{Na}_3\text{H}[\text{Pt}^{\text{IV}}\text{W}_6\text{O}_{24}\text{H}_3] \cdot \text{CH}_3\text{CN}$

Furthermore, the capillary electrophoresis have also been used to monitor this process (see Supplementary Figure 8). The retention time of $\text{PtW}_6\text{O}_{24}$ before HER and its re-oxidized species by Br_2 after HER almost unchanged, further confirming that $\text{PtW}_6\text{O}_{24}$ compound is a reversible redox-active catalyst.

Supplementary Figure 8 | Electrochromatogram for 0.25 mM of $\text{PtW}_6\text{O}_{24}$ compound in a 20 mM $\text{NaH}_2\text{PO}_4\text{-H}_3\text{PO}_4$ buffer (pH=3) before HER (a), and its re-oxidized species by Br_2 after HER (b). The retention time of $\text{PtW}_6\text{O}_{24}$ before HER and its re-oxidized species by Br_2 after HER almost unchanged, further confirming that $\text{PtW}_6\text{O}_{24}$ compound is a reversible redox-active catalyst.

We made the modification in the supporting information as shown below.

Supplementary Figure 7 | The electrospray-ionization mass spectra (EIS) of $\text{PtW}_6\text{O}_{24}$ compound before and after HER in 0.5 M H_2SO_4 aqueous solution as well as its re-oxidized species by Br_2 . The test samples are prepared by precipitating the polyoxoanion with tetrabutylammonium (TBA) bromide and dissolved in CH_3CN . (a) Before HER, the signals of $m/z = 1689$ can be assigned to $\text{H}_4[\text{Pt}^{\text{IV}}\text{W}_6\text{O}_{24}\text{H}_3]^-$ species. (b) After $\text{PtW}_6\text{O}_{24}$ catalyst has undergone HER for 10 cycles in 0.5 M H_2SO_4 aqueous solution, the signals of $m/z = 1690$, $m/z = 1795$, $m/z = 1898$, and $m/z = 1972$ can be assigned to $\text{H}_4[\text{Pt}^{\text{II}}\text{W}_6\text{O}_{24}\text{H}_5]^-$, $\text{NaH}_3[\text{Pt}^{\text{II}}\text{W}_6\text{O}_{24}\text{H}_5]^- \cdot (\text{CH}_3\text{CN})_2$, $\text{Na}_4[\text{Pt}^{\text{II}}\text{W}_6\text{O}_{24}\text{H}_5]^- \cdot (\text{CH}_3\text{CN})_2(\text{H}_2\text{O})_2$, and $(\text{TBA})\text{NaH}_2[\text{Pt}^{\text{II}}\text{W}_6\text{O}_{24}\text{H}_5]^- \cdot \text{CH}_3\text{CN}$ species, respectively. (c) When the molecular catalyst was re-oxidized by Br_2 in above solution, the signals of $m/z = 1710$, $m/z = 1732$, $m/z = 1753$, $m/z = 1777$

and $m/z = 1798$ can be assigned to $\text{NaH}_3[\text{Pt}^{\text{IV}}\text{W}_6\text{O}_{24}\text{H}_3]^-$, $\text{Na}_2\text{H}_2[\text{Pt}^{\text{IV}}\text{W}_6\text{O}_{24}\text{H}_3]^-$, $\text{NaH}_3[\text{Pt}^{\text{IV}}\text{W}_6\text{O}_{24}\text{H}_3]^- \cdot \text{CH}_3\text{CN}$, $\text{Na}_4[\text{Pt}^{\text{IV}}\text{W}_6\text{O}_{24}\text{H}_3]^-$, and $\text{Na}_3\text{H}[\text{Pt}^{\text{IV}}\text{W}_6\text{O}_{24}\text{H}_3]^- \cdot \text{CH}_3\text{CN}$, respectively. The above experiments demonstrate that $\text{PtW}_6\text{O}_{24}$ compound behaves as a reversible redox-active catalyst. (see Supplementary Information page 4)

Supplementary Figure 8 | Electrochromatogram for 0.25 mM of $\text{PtW}_6\text{O}_{24}$ compound in a 20 mM $\text{NaH}_2\text{PO}_4\text{-H}_3\text{PO}_4$ buffer (pH=3) before HER (a), and its re-oxidized species by Br_2 after HER (b). The retention time of $\text{PtW}_6\text{O}_{24}$ before HER and its re-oxidized species by Br_2 after HER almost unchanged, further confirming that $\text{PtW}_6\text{O}_{24}$ compound is a reversible redox-active catalyst. (see Supplementary Information pages 4-5)

Reviewer 2

Revision after more data is provided.

This reviewer would like to get access to the DFT computed geometries and frequencies of all species included in the manuscript, particularly those corresponding to the Tafel-like H-H forming transition state. The barrier reported is very low, and has to be verified. Normally, this data is included in the Supplementary Information section, but it is missing here. Alternatively, the authors may upload input and output files to any digital repository and provide links to access the data.

Minor questions:

The version of the program used needs to be properly mentioned, as well as properly cited.

Our response:

Now a data set of computational results in the manuscript is available in the ioChem-BD repository and can be accessed via <https://doi.org/10.19061/iochem-bd-6-27>. The optimized geometries are also provided as the Supplementary Information. The reviewer could check the Tafel-like H-H forming transition state from the above link which named as **TS-4e-4H**. The transition states (TS) have been confirmed by the existence of only one imaginary frequency along the reaction coordinate and intrinsic reaction coordinates (IRC) calculations which indeed connect the right reactants and products (Supplementary Figure S6). Finally, the right version of the program has been included and cited in the section of method “DFT computational details”.

We made the corresponding modification in the supporting information as shown below.

Supplementary Figure 56 | Intrinsic reaction coordinate (IRC) between $[\text{H}_6\text{PtW}_6\text{O}_{24}]^{4e-/4\text{H}(\text{Pt})}$ and $\text{H}_2+[\text{H}_6\text{PtW}_6\text{O}_{24}]^{2e-/2\text{H}}$ through the transition states (TS) which were obtained by using UM06/6-31G(d,p) level of theory.(see Supplementary Information page 32)

We also made the corresponding modification in the manuscript as shown below.

DFT computational details. All calculations were performed through the facilities provided by the Gaussian09 package (revision D.01)⁴¹. Geometry optimizations for all intermediates and transition states were carried out at the M06 level without symmetry restrictions⁴². The LANL2DZ basis set was employed for the Pt and W, whereas the 6-31G(d, p) basis set was used for the O and H^{43,44}. To confirm the stability of all structures, frequency calculations were performed at the same level as optimization. The transition states (TS) were confirmed by the existence of only one imaginary frequency along the reaction coordinate and intrinsic reaction coordinates (IRC) calculations which indeed connect the right reactants and products (Supplementary Fig. S56)⁴⁵. The solvation effects of water were introduced by using the PCM model⁴⁶. Furthermore, the single-point energies of all stationary points were completed at (U)M06/PCM(H₂O)/[6-311++G(d,p)/SDD(Pt&W)] level for all energy calculations⁴⁷. Finally, a data set of computational results is available in the ioChem-BD repository and can be accessed via <https://doi.org/10.19061/iochem-bd-6-27> (<http://www.iochembd.org/>)⁴⁸. (see manuscript page 20)

Reviewer 3

The manuscript submitted by Li et al report on the remarkable HER efficiency of two platinum-containing polyoxometalates (POMs), abbreviated $\text{PtW}_6\text{O}_{24}$ and $\text{Pt}_2(\text{W}_5\text{O}_{18})_2$. In a general point of view, the presented work appears well-done and highly convincing. The authors demonstrate that containing-platinum POMs as soluble oxide analogues are able to mimic the platinum oxide behavior toward hydrogen evolution giving highly efficient HER catalysts. The manuscript proposes a multi-scale characterization of the catalyst using electrochemistry, XPS, X-ray absorption spectroscopy (XANES and EXAFS), TEM and STEM, vibrational spectroscopies (Raman and infrared, impedance spectroscopy.... Furthermore, experimental data were supported by theoretical calculation at the DFT level, thus giving a set of reliable arguments consistent with the hypotheses. Undoubtedly, these two Pt-containing POMs are stable, highly processable, very efficient and resistant against usual Pt-contaminant. This submitted manuscript could display the requested scientific quality to be published in NatureCOMM. The first critical point of the MS is certainly the repeated comparison of the HER performances of Pt-POM/catalysts with those of the commercial 20% Pt/C. The sentence “better than that of commercial 20% Pt/C” appears at least twelve times in the text. The reader understand that the commercial 20% Pt/C is a very bad HER catalyst, with not optimized Pt-dispersion and probably containing platinum-contaminant agents. Furthermore, the procedures of preparation of both type of catalyst are necessarily different. For instance, the nature of the used carbon is different, such as Ketjen carbon for Pt-POM/catalysts leading to important changes of the surface area and conductivity. Actually, this is easy to get a poorly HER active platinum-based catalyst.

This point must be corrected and the highest HER performances of the two Pt-POMs/C should be commented objectively and scientifically.

Furthermore, the paper is divided in two redundancy part dealing with the HER properties of the $\text{PtW}_6\text{O}_{24}/\text{C}$, and those of the $\text{Pt}_2(\text{W}_5\text{O}_{18})_2/\text{C}$, respectively. The first part appears very interesting, dealing with the electrochemical behavior of the Pt(IV) center embedded within the $\{\text{W}_6\text{O}_{24}\}$ framework. Calculations are consistent with the expected quasi-square Pt(II) center, while the HER process resulted from the presence of Pt(I)-H hydrid group. The second part ($\text{Pt}_2(\text{W}_5\text{O}_{18})_2/\text{C}$) brings nothing new and should be deleted. The authors did not give any arguments about the selection of this $\text{Pt}_2(\text{W}_5\text{O}_{18})_2$ POM species.

Less minor point: In the “Methods” section, the synthetic procedure of the two Pt-POMs/C, given in the text is not consistent with the scheme 1 (supp Info). In the text Nafion is used while in the supp Info, Melamine-Formaldehyde is added. This should be cleared.

In conclusion, this paper should disserve publication in NatureCOMM, therefore some revisions are still needed.

Our response:

We do thank your positive comments.

Comment 1: The first critical point of the MS is certainly the repeated comparison of the HER performances of Pt-POM/catalysts with those of the commercial 20% Pt/C. The sentence “better than that of commercial 20% Pt/C” appears at least twelve times in the text. The reader understand that the commercial 20% Pt/C is a very bad HER catalyst, with not optimized Pt-dispersion and probably containing platinum-contaminant agents. Furthermore, the procedures of preparation of both type of catalyst are necessarily different. For instance, the nature of the used carbon is different, such as Ketjen carbon for Pt-POM/catalysts leading to important changes of the surface area and conductivity. Actually, this is easy to get a poorly HER active platinum-based catalyst. This point must be corrected and the highest HER performances of the two Pt-POMs/C should be commented objectively and scientifically.

Our response:

We agree with the reviewer's comments. Although commercial 20% Pt/C is widely used as the reference electrocatalyst for studying HER reactions, its catalytic activity is already not the best as the reviewer commented. No matter in size, morphology, or dispersion of metal Pt-based electrocatalysts on various loading materials, there still has enough space to improve its catalytic activity, which has been proved by a number of literature reports in recent five years^{9-14, 39, 40}. It also becomes an important driving force for scientists to continue developing new Pt-based electrocatalysts. Therefore, “*PtW₆ and Pt₂W₁₀ are more active than commercial Pt/C*”, which was over emphasized in our manuscript, should just be an experimental fact but not an unusual concern. We agree with the reviewer’s comment and correct the relevant statement in the revised manuscript. The main contribution of this work is to reveal the critical role of O atom in the oxidized platinum-based catalysts, further complement the knowledge boundary of Pt-based electrocatalytic HER, and may provide a new way to design more efficient and lower content Pt-based catalysts for HER.

We made the modification in the manuscript as shown below.

The mass activity and specific activity were normalized by the mass loading and the ECSA of Pt. As depicted in Fig. 2c, at an overpotential of 77 mV, PtW₆O₂₄/C displays a mass activity of 20.175 A mg⁻¹, while the mass activity of 20% Pt/C is 0.398 A mg⁻¹. Furthermore, 1% PtW₆O₂₄/C displays a specific activity of 35.266 mA cm⁻² at 50 mV, and the value of 20% Pt/C is 0.132 mA cm⁻² under the same condition. (see manuscript page 9)

Herein, it should be also clarified that although commercial Pt/C is widely used as a standard reference for HER research, its performance is already not the best. No matter in size, morphology, and dispersion of metal Pt, there exists enough space to improve its catalytic activity^{9-14,39,40}. Thus, surpassing commercial Pt/C does not mean that metal Pt-based catalysts are out of date, which exactly suggests an important driving force for deeply developing such state-of-the-art catalysts. (see manuscript page 16)

The electrochemical experiments show that PtW₆O₂₄/C and Pt₂(W₅O₁₈)₂/C possess the overpotentials of 22 mV and 26 mV at a current density of 10 mA cm⁻², and their mass activities are 20.175 A mg⁻¹ and 10.976 A mg⁻¹, respectively. (see manuscript page 16)

Therefore, Pt-O can be utilized as a new active site towards HER. This work answers the

important role of O atoms in the oxidized platinum-based electrocatalytic HER, which may bring new enlightenment for the design and preparation of more efficient Pt-based electrocatalysts in the near future. (see manuscript pages 16-17)

Reference (see manuscript pages 22, 25)

9. Li, M. *et al.* Ultrafine jagged platinum nanowires enable ultrahigh mass activity for the oxygen reduction reaction. *Science* **354**, 1414-1419 (2016).
10. Chen, G. *et al.* Interfacial electronic effects control the reaction selectivity of platinum catalysts. *Nat. Mater.* **15**, 564-569 (2016).
11. Lin, L. *et al.* Low-temperature hydrogen production from water and methanol using Pt/ α -MoC catalysts. *Nature* **544**, 80-83(2017).
12. Wang, S. *et al.* Ultrafine Pt nanoclusters confined in a calixarene-based $\{Ni_{24}\}$ coordination cage for high-efficient hydrogen evolution reaction. *J. Am. Chem. Soc.* **138**, 16236-16239 (2016).
13. Bu, L. *et al.* Biaxially strained PtPb/Pt core/shell nanoplate boosts oxygen reduction catalysis. *Science* **354**, 1410-1414 (2016).
14. Wang, A. *et al.* Heterogeneous single-atom catalysis. *Nat. Rev. Chem.* **2**, 65-81 (2018).
39. Wang, P. *et al.* Precise tuning in platinum-nickel/nickel sulfide interface nanowires for hydrogen evolution catalysis. *Nat. Commun.* **8**, 14580 (2017).
40. Zhang, Z. *et al.* Crystal phase and architecture engineering of lotus-thalamus-shaped Pt-Ni anisotropic superstructures for hydrogen efficient electrochemical hydrogen evolution. *Adv. Mater.* **30**, 1801741 (2018).

Comment 2: Furthermore, the paper is divided in two redundancy part dealing with the HER properties of the PtW_6O_{24}/C , and those of the $Pt_2(W_5O_{18})_2/C$, respectively. The first part appears very interesting, dealing with the electrochemical behavior of the Pt(IV) center embedded within the $\{W_6O_{24}\}$ framework. Calculations are consistent with the expected quasi-square Pt(II) center, while the HER process resulted from the presence of Pt(I)-H hydrid group. The second part ($Pt_2(W_5O_{18})_2/C$) brings nothing new and should be deleted. The authors did not give any arguments about the selection of this $Pt_2(W_5O_{18})_2$ POM species.

Our response:

According to the suggestion from Reviewer and Senior Editor (“*We wish to note that Reviewer #3 requests you remove the portion on $Pt_2(W_5O_{18})_2$ from the manuscript. However, we believe that this work is still important. We suggest moving this portion into the Supplementary Information in an effort to focus the main text on the first POM discussed*”), we removed the second part ($Pt_2(W_5O_{18})_2/C$) into Supplementary Information with yellow highlight. The reason that we still select $Pt_2(W_5O_{18})_2$ has also been proposed in the manuscript. “Based on the DFT and XAS results, $[H_6Pt(II)W_6O_{24}]^{2e/2H}$ may represent an important intermediate for the high HER performance of $H_6PtW_6O_{24}$. This result aroused our curiosity to detect the electrocatalytic activity of another polyoxometalate $Na_3K_5[Pt(II)_2(W_5O_{18})_2]$ ($Pt_2(W_5O_{18})_2$) since it contains a similar Pt^{II}-O moiety in the molecular structure. The overpotential of 1% $Pt_2(W_5O_{18})_2/C$ is 22 mV at 10 mA cm⁻². Its exchange current density and mass activity at 77 mV are 1.65 mA cm⁻² and 20.175 A mg⁻¹, respectively, which

are quite close to $\text{PtW}_6\text{O}_{24}$. More detailed data is provided in the Supplementary (Supplementary Fig. 36-55)³⁸. This result further confirms that Pt-O bond is the active site during the HER process.”

We made the corresponding modification in supporting information as shown below.

We further explore the HER performance of electrocatalyst $\text{Pt}_2(\text{W}_5\text{O}_{18})_2/\text{C}$. As shown in Supplementary Figure. 42a, the $[\text{Pt}_2(\text{W}_5\text{O}_{18})_2]^{8-}$ anion is composed of two Pt(II) ions capped by two lacunary Lindqvist structures ($\text{W}_5\text{O}_{18}^{6-}$). Each Pt(II) coordinates with the terminal oxygen atoms of W_5O_{18} fragment in a square plane environment. The distance of Pt-O bond varies within the scope of 1.984(6)-2.000(6) Å. The distance between two Pt atoms is 3.1315(8) Å, which is obviously longer than the metal-metal bond distance. The 1% $\text{Pt}_2(\text{W}_5\text{O}_{18})_2/\text{C}$ also exhibits an excellent electrocatalytic hydrogen evolution performance with an overpotential of 26 mV at 10 mA cm^{-2} (Supplementary Figure 42b and Figure 43), which is similar to that of 1% $\text{PtW}_6\text{O}_{24}/\text{C}$. The Tafel slope is 29.8 mV dec^{-1} with the Volmer-Tafel mechanism (Supplementary Figure 42c and Figure 44). The exchange current density is 1.42 mA cm^{-2} and the mass activity is 10.976 mg^{-1} at 77 mV, respectively. The TOFs at 100 mV is 16.63 s^{-1} . All these results are similar to 1% $\text{PtW}_6\text{O}_{24}/\text{C}$ (Supplementary Figure 42b and Figures 45-48). Electrochemical impedance spectroscopy (EIS) (Supplementary Figure 49) suggests the charge transfer resistance of 1% $\text{Pt}_2(\text{W}_5\text{O}_{18})_2/\text{C}$, implying a fast HER kinetics. In addition, 1% $\text{Pt}_2(\text{W}_5\text{O}_{18})_2/\text{C}$ also possesses an excellent FE (nearly 100%) (Supplementary Figure 50), good anti-toxicity (Supplementary Figure 51) and stability during the whole HER process (Supplementary Figure 42d, Figures. 52-55 and Tables 4-5). These observations further confirm that Pt-O bond can be a more active site than metallic Pt0 toward electrocatalytic HER. (see Supplementary Information page 25)

Supplementary Figure 42 | Structure and HER performance of the $\text{Pt}_2(\text{W}_5\text{O}_{18})_2/\text{C}$ catalyst. (a) The ball and stick representation of $\text{Pt}_2(\text{W}_5\text{O}_{18})_2$. (b) The polarization curves of 1% $\text{Pt}_2(\text{W}_5\text{O}_{18})_2/\text{C}$ and 20% Pt/C at a current density of 10 mA cm^{-2} in N_2 -saturated $0.5 \text{ M H}_2\text{SO}_4$. Inset: mass activity of 1% $\text{Pt}_2(\text{W}_5\text{O}_{18})_2/\text{C}$ and 20% Pt/C at 77 mV . (c) Tafel slope of 1% $\text{Pt}_2(\text{W}_5\text{O}_{18})_2/\text{C}$ and 20% Pt/C. (d) Time-dependent current density current of 1% $\text{Pt}_2(\text{W}_5\text{O}_{18})_2/\text{C}$ and 20% Pt/C within 24 h.

We also made the corresponding modification in the manuscript as shown below.

Based on the DFT and XAS results, $[\text{H}_6\text{Pt}^{\text{II}}\text{W}_6\text{O}_{24}]^{2e/2\text{H}}$ may represent an important intermediate for the high HER performance of $\text{H}_6\text{PtW}_6\text{O}_{24}$. This result aroused our curiosity to detect the electrocatalytic activity of another polyoxometalate $\text{Na}_3\text{K}_5[\text{Pt}^{\text{II}}_2(\text{W}_5\text{O}_{18})_2]$ ($\text{Pt}_2(\text{W}_5\text{O}_{18})_2$) since it contains a similar $\text{Pt}^{\text{II}}\text{-O}$ moiety in the molecular structure. The overpotential of 1% $\text{Pt}_2(\text{W}_5\text{O}_{18})_2/\text{C}$ is 22 mV at 10 mA cm^{-2} . Its exchange current density and mass activity at 77 mV are 1.65 mA cm^{-2} and 20.175 A mg^{-1} , respectively, which are quite close to $\text{PtW}_6\text{O}_{24}$. More detailed data is provided in the Supplementary (Supplementary Fig. 36-55)³⁸. This result further confirms that Pt-O bond is the active site during the HER process. (see manuscript pages 15-16)

Comment 3: Less minor point: In the “Methods” section, the synthetic procedure of the two Pt-POMs/C, given in the text is not consistent with the scheme 1 (supp Info). In the text Nafion is used while in the supp Info, Melamine-Formaldehyde is added. This should be cleared.

Our response:

We are sorry for this carelessness. In the preparation of Pt-POMs/C, Melamine-Formaldehyde was used instead of Nafion. Therefore, the error in the method has been corrected and marked with yellow highlight.

We made the corresponding modification in the manuscript as shown below.

Synthesis of $\text{PtW}_6\text{O}_{24}/\text{C}$. 0.065 g crystal of $\text{Na}_5[\text{H}_3\text{Pt}(\text{IV})\text{W}_6\text{O}_{24}]$ was uniformly dispersed in $1 \text{ mL H}_2\text{O}$, and 5 mg Ketjen black carbon was added to, stirring at room temperature for 2 hours. Then $10 \mu\text{L}$ Melamine-Formaldehyde was added to the aqueous and stirred 4 hours. The electrocatalyst can be obtained after centrifuged and dried. The obtained sample is denoted as $\text{PtW}_6\text{O}_{24}/\text{C}$.

Synthesis of $\text{Pt}_2(\text{W}_5\text{O}_{18})_2/\text{C}$. 0.065 g crystal of $\text{Na}_3\text{K}_5[\text{Pt}(\text{II})_2(\text{W}_5\text{O}_{18})_2]$ was uniformly dispersed in $1 \text{ mL H}_2\text{O}$, and 5 mg Ketjen black carbon was added to, stirring at room temperature for 2 hours. Then $10 \mu\text{L}$ Melamine-Formaldehyde was added to the aqueous and stirred 4 hours. The electrocatalyst can be obtained after centrifuged and dried. The obtained sample is denoted as $\text{Pt}_2(\text{W}_5\text{O}_{18})_2/\text{C}$. (see manuscript page 18)

REVIEWERS' COMMENTS:

Reviewer #1 (Remarks to the Author):

As noted in review of the original manuscript, the key finding is that the partial reduction of Pt cations bound via oxide ligands to reduced W ions within molecular heteropolytungstate clusters are more effective than Pt(0), and comparable to some of the most effective reported in the literature in the hydrogen evolution reaction. Importantly, the W-based clusters appear to serve as stable, redox catalysts, consistent with reversible reduction / reoxidation under turnover conditions. This latter point is sufficiently important that additional experimental support for this conclusion was requested for inclusion in the revised manuscript. The authors have now included this additional data, and it indeed makes a strong case for catalyst stability under turnover conditions. As such, publication of the revised manuscript is now recommended (with compliments to the authors on the quality of their newly added experiments and data). Ira A. Weinstock

Reviewer #2 (Remarks to the Author):

This reviewer acknowledges author's revision.
In particular, I'd like to congratulate them for the way that they have made available the DFT data through such ioChem-BD repository. This is really a nice form of publishing computational chemistry results.

Publish as it is.

Reviewer #3 (Remarks to the Author):

The authors fulfill all the recommendations required by the three referees. This is a nice paper which corresponds to the high standard of Nature Communication.

Response to reviewers' comments

Reviewer #1

As noted in review of the original manuscript, the key finding is that the partial reduction of Pt cations bound via oxide ligands to reduced W ions within molecular heteropolytungstate clusters are more effective than Pt(0), and comparable to some of the most effective reported in the literature in the hydrogen evolution reaction. Importantly, the W-based clusters appear to serve as stable, redox catalysts, consistent with reversible reduction / reoxidation under turnover conditions. This latter point is sufficiently important that additional experimental support for this conclusion was requested for inclusion in the revised manuscript. The authors have now included this additional data, and it indeed makes a strong case for catalyst stability under turnover conditions. As such, publication of the revised manuscript is now recommended (with compliments to the authors on the quality of their newly added experiments and data). Ira A. Weinstock.

Our response:

Thank the reviewer for reviewing our manuscript, and the recognition of our work.

Reviewer #2

This reviewer acknowledges author's revision.

In particular, I'd like to congratulate them for the way that they have made available the DFT data through such ioChem-BD repository. This is really a nice form of publishing computational chemistry results.

Publish as it is.

Our response:

Thank the reviewer for reviewing our manuscript, and the recognition of our work.

Reviewer #3

The authors fulfill all the recommendations required by the three referees. This is a nice paper which corresponds to the high standard of Nature Communication.

Our response:

Thank the reviewer for reviewing our manuscript, and the recognition of our work.